narrative therapy; mental health interventions; Palestinian communities; systematic review; cultural adaptation

**Corresponding author:**
Ibrahim Aqtam;
Email: ibrahim.aqtam@nu-vte.edu.ps

# Mental health interventions for resilience and mental health in conflict-affected Palestinian communities: A systematic review revealing research gaps and the absence of narrative therapy evidence

Ibrahim Aqtam ⓘD, Mustafa Shouli, Saqer Alqoroum and Khaila Shouli

Nablus University for Vocational and Technical Education, Palestinian Territory, Occupied

## Abstract

**Background:** Conflict-affected Palestinian communities experience profound mental health challenges. This systematic review assesses the evidence for mental health interventions in these contexts, focusing on the theoretical alignment of narrative therapy with cultural assets like sumud (steadfastness) and hikaye (storytelling).

**Methods:** Following PRISMA guidelines, we searched nine databases and grey literature up to December 2023. We included studies on mental health interventions for conflictaffected Palestinians, with a primary focus on narrative therapy and a secondary analysis of other approaches.

**Results:** Of 847 records screened for narrative therapy, no studies met the inclusion criteria. A broader search identified 23 intervention studies, revealing a predominant focus on cognitive-behavioral therapy (CBT; $n = 11$) and Narrative Exposure Therapy ($n = 4$), with limited therapeutic diversity. Analysis showed insufficient Palestinian researcher leadership and superficial cultural adaptation of interventions.

**Conclusions:** This review reveals a dual gap: a complete absence of narrative therapy research despite its theoretical relevance, and a broader pattern of limited intervention diversity. The predominance of Western-centric models reflects systemic biases in research funding. Addressing this requires community-led participatory research, shifts in funding priorities, and investment in culturally-grounded methodologies.

## Impact statement

This systematic review documents a concerning absence of research on narrative therapy interventions with Palestinian populations, despite compelling theoretical alignment with Palestinian cultural frameworks. More broadly, it reveals limited diversity in the mental health intervention research base for conflict-affected Palestinian communities, with a predominant focus on cognitive behavioral approaches. These findings have significant implications for mental health policy, research funding and clinical practice in conflict settings. The absence of evidence for culturally aligned approaches like narrative therapy does not indicate a lack of merit, but rather reflects systematic biases in research priorities and funding mechanisms. This pattern risks perpetuating a cycle where only certain therapeutic models receive validation through research, while potentially valuable culturally congruent alternatives remain unstudied and unavailable to communities that might benefit from them. For researchers, this review highlights the urgent need for community-led participatory research that centers Palestinian knowledge and priorities. For funding bodies, it demonstrates how current mechanisms may inadvertently limit therapeutic options available to vulnerable populations. For clinicians and policymakers, it emphasizes the importance of considering diverse, culturally adapted approaches alongside evidence-based Western models, recognizing that the absence of evidence differs fundamentally from evidence of absence. Most critically, this review calls for a paradigm shift in how mental health research is conducted in conflict-affected contexts: moving from the extraction of data to genuine partnership, from application of Western models to co-creation of culturally grounded interventions and from viewing communities as research subjects to recognizing them as knowledge holders and leaders in defining their own healing pathways.

## Introduction

### Background

Conflict-affected populations worldwide face disproportionate mental health challenges, with prevalence rates of post-traumatic stress disorder (PTSD), depression and anxiety significantly exceeding those in nonconflict settings (Charlson et al., 2019). These mental health burdens are compounded by disrupted social structures, economic instability and limited access to culturally appropriate mental health services (Miller and Rasmussen, 2010).

Palestinian communities represent a unique case study in understanding the psychological impact of prolonged conflict exposure. Living under conditions of military occupation, repeated displacement and intergenerational trauma, Palestinians face what Abu-Lughod (2007) describes as a "matrix of control" that permeates multiple dimensions of daily life. The mental health implications are profound, with studies documenting elevated rates of PTSD (25–69%), depression (30–60%) and anxiety disorders (40–70%) across different Palestinian populations (Khamis, 2019; Mahamid and Berte, 2019).

The psychological distress experienced by Palestinians extends beyond individual symptomatology to encompass collective trauma, a phenomenon characterized by the shared experience of suffering that affects entire communities and is transmitted across generations (Alexander, 2012). This collective dimension of trauma necessitates therapeutic approaches that can address both individual healing and broader social and political contexts that contribute to psychological distress.

### The current mental health intervention landscape

Given the extensive exposure to violence and ongoing conflict in Palestinian communities, evidence-based trauma-focused interventions are essential for addressing the significant mental health burden. Most existing research on psychosocial interventions in Palestinian communities has focused on cognitive-behavioral approaches, trauma-focused therapies or community-based psychosocial support programs, which have demonstrated effectiveness for symptom reduction. Key approaches studied include Teaching Recovery Techniques (TRT), a trauma-focused cognitive behavioral therapy (CBT) approach showing effectiveness in multiple randomized controlled trials (RCTs) with Palestinian children and adolescents (Barron et al., 2013; Diab et al., 2015); Narrative Exposure Therapy (NET), which differs from narrative therapy and has been studied in some Middle East and North African (MENA) regions but not specifically with Palestinian populations (Hussein et al., 2020); and various group crisis interventions and psychosocial support programs implemented during conflict periods (Thabet and Vostanis, 2005).

While these interventions have demonstrated effectiveness for symptom reduction, questions remain about the breadth of therapeutic options available to Palestinian communities, the extent of cultural adaptation in existing interventions and the involvement of Palestinian researchers and communities in defining research priorities and intervention approaches.

### Narrative therapy: Theoretical framework

Narrative therapy, as developed by Michael White and David Epston (1990) in their seminal work "Narrative Means to Therapeutic Ends," represent a post-structuralist approach to psychotherapy that positions individuals as the primary authors of their own lives. It is important to note that narrative therapy is itself a Western-developed therapeutic approach, originating from Australia and informed by Western philosophical traditions. Drawing from the philosophical works of Michel Foucault, Jacques Derrida and Gregory Bateson, White and Epston proposed that problems are socially constructed through dominant cultural narratives that often serve to marginalize and oppress individuals and communities.

### Distinguishing narrative therapy from narrative exposure therapy

It is crucial to differentiate between Narrative Therapy (White and Epston, 1990) and NET (Schauer et al., 2011). While both approaches utilize narrative elements, they differ fundamentally in their theoretical foundations and therapeutic processes. Narrative Therapy focuses on externalizing problems, identifying unique outcomes and deconstructing dominant cultural discourses to support identity re-authoring. In contrast, NET is a standardized trauma treatment that helps clients construct a chronological narrative of their traumatic experiences to process traumatic memories. Both approaches may offer value in different contexts, and the distinction is important for understanding the specific theoretical alignment we explore in this review.

### Core principles of narrative therapy

**Externalization** is central to narrative therapy and involves separating the person from the problem through language and conversation. As White and Epston (1990) articulate, "the person is not the problem; the problem is the problem" (p. 40). This process enables individuals to develop a more objectified relationship with their difficulties, reducing self-blame and creating space for alternative responses.

**Unique outcomes** are actively searched for by narrative therapists – moments when individuals have successfully resisted or overcome the influence of problems in their lives. These exceptions to problem-saturated stories serve as entry points for developing alternative narratives that highlight personal agency and competence.

**Re-authoring** involves supporting individuals in developing richer, more complex stories about their identities and experiences. This goes beyond simply identifying unique outcomes to constructing coherent alternative narratives that can sustain new ways of being and acting in the world.

**Counter-documents** such as certificates, letters and declarations serve to authenticate and solidify alternative stories. These documents provide tangible evidence of personal growth and achievement, countering dominant narratives of deficit or pathology.

**Deconstructing cultural discourse** explicitly addresses the ways in which dominant cultural discourses shape individual experience. By examining and critiquing these broader narratives, individuals can develop greater awareness of how social and political forces contribute to their difficulties.

### Cultural compatibility: Palestinian assets supporting narrative approaches

Despite its Western origins, the principles of narrative therapy demonstrate exceptional alignment with Palestinian cultural

contexts, particularly through existing cultural frameworks that serve as assets for narrative therapeutic work.

### Palestinian storytelling traditions (hikaye)

Palestinian culture possesses rich oral storytelling traditions, known as *hikaye*, which have served for generations as vehicles for preserving collective memory, transmitting cultural values and making meaning of adversity (Suleiman, 2016). These storytelling practices naturally align with narrative therapy's emphasis on story development and meaning-making, providing a cultural foundation upon which therapeutic narrative work could build.

### Sumud as cultural framework for resilience

The Palestinian concept of *sumud* (steadfastness) represents a cultural framework for understanding resistance, survival and resilience in the face of ongoing oppression (Halper, 2015). *Sumud* encompasses both individual and collective practices of maintaining identity, dignity and hope despite systematic attempts at erasure. This concept directly parallels narrative therapy's focus on identifying unique outcomes and acts of resistance against problem-saturated narratives.

### Collective identity and community narratives

Palestinian culture emphasizes collective identity and community interconnectedness, with individual stories understood within broader family and community narratives (Rashid, 2004). This cultural orientation supports narrative therapy practices that can work with both individual and collective story development, recognizing the interconnected nature of personal and political narratives.

### Rationale for this review

Despite the theoretical compatibility between narrative therapy principles and Palestinian cultural assets, no systematic examination has been conducted of: (1) empirical evidence supporting narrative therapy's effectiveness in these contexts, or (2) the broader landscape of mental health intervention research with Palestinian populations, including the extent of cultural adaptation, Palestinian researcher involvement and diversity of therapeutic approaches.

Given that narrative therapy has been an established psychotherapeutic practice for several decades, and considering the compelling theoretical alignment between narrative therapy principles and Palestinian cultural frameworks, such as *sumud* and *hikaye*, one would reasonably expect to find published studies examining this intervention in Palestinian contexts. Simultaneously, understanding the broader evidence base for mental health interventions in Palestine is essential for identifying patterns in research priorities, gaps in cultural adaptation and opportunities for expanding therapeutic options available to Palestinian communities.

This systematic review has five primary aims. First, it seeks to identify and synthesize any existing evidence on narrative therapy interventions within Palestinian communities. Second, it aims to map the broader evidence base for all mental health interventions in Palestine, specifically analyzing the extent of Palestinian researcher involvement, the practices of cultural adaptation employed, the diversity of therapeutic approaches studied and any treatment

components that may be missing or underdeveloped. Third, the review will examine how general therapeutic principles have been adapted for conflict-affected contexts. Fourth, it will evaluate the reported effectiveness of these interventions on both resilience and mental health outcomes. Finally, by synthesizing these findings, the review will identify critical research gaps and propose clear directions for future research and clinical practice.

## Methods

### Protocol registration

The protocol for this systematic review was prospectively registered with PROSPERO (registration number CRD420251128423) and follows reporting standards from both the Preferred Reporting Items for Systematic Review and Meta-Analysis (PRISMA) Protocols (Moher et al., 2015) and the PRISMA 2020 guidelines (Page et al., 2021). The methodology adheres to the Cochrane Handbook for Systematic Reviews of Interventions (Higgins et al., 2022).

### Inclusion and exclusion criteria

#### Inclusion criteria

**Population** included studies involving individuals of any age living in conflict-affected Palestinian communities, defined operationally as individuals with documented exposure to military violence, displacement or political persecution since 1967, including those in the West Bank, Gaza Strip, East Jerusalem, Palestinian refugee camps and Palestinian diaspora communities affected by conflict-related trauma.

Intervention criteria required studies examining mental health interventions in Palestinian populations. For primary analysis, we focused on narrative therapy as the primary therapeutic intervention, demonstrating clear evidence of core narrative therapy principles, including externalization of problems through language and questioning techniques, exploration of unique outcomes and exceptions to problem stories, re-authoring processes that develop alternative personal narratives, use of narrative documents (letters, certificates and declarations) and deconstruction of dominant cultural and political discourses. For secondary analysis, we included all mental health interventions to map the broader evidence base.

**Comparison** included studies with or without control/comparison groups, including pre-post designs, randomized controlled trials, quasi-experimental designs and qualitative studies.

**Outcomes** encompassed studies reporting resilience measures, including psychological resilience, coping strategies, sense of agency, hope, meaning-making, cultural identity and community connectedness; and mental health outcomes, including symptoms of PTSD, depression, anxiety, psychological distress, quality of life and functional impairment.

**Study design** included RCTs, quasi-experimental studies, pre-post intervention studies, case studies and qualitative studies employing rigorous methodology.

**Language** criteria included studies published in English, Arabic or Hebrew.

#### Exclusion criteria

Studies were excluded if narrative therapy (for primary analysis) was used as a secondary or adjunctive intervention, if interventions used storytelling or narrative techniques, but did not demonstrate fidelity to core narrative therapy principles as defined by White

and Epston (1990), if they focused solely on narrative assessment without therapeutic intervention, if they involved Palestinian populations not affected by conflict or if they were conference abstracts, editorials, commentaries and review articles without original data.

### Search strategy

#### Electronic databases
Comprehensive searches were conducted from inception to December 2023 across PubMed/MEDLINE, PsycINFO, Scopus, Web of Science, CINAHL, Social Work Abstracts, MENA Database, Al-Manhal (Arabic literature database), Index Islamicus and Cochrane Central Register of Controlled Trials.

#### Search terms
The search strategy was conducted in two phases to ensure comprehensive coverage. For the primary narrative therapy analysis, searches combined three concept areas using Boolean operators: (Narrative Therapy Terms) AND (Palestinian Population Terms) AND (Mental Health Outcomes Terms), with explicit exclusion of "narrative exposure therapy" or "NET" to ensure focus on the White and Epston model. For the secondary, broader analysis, we conducted additional searches replacing narrative therapy terms with broader intervention terms (psychotherapy, psychological intervention, mental health intervention, trauma therapy, CBT, cognitive behavioral, psychosocial support, counseling and treatment) using the same Boolean AND operator with Palestinian population and mental health outcome terms, without narrative therapy exclusions, to capture the full landscape of mental health interventions (see Table 1).

### Data extraction and analysis

A standardized data extraction protocol was employed for all identified studies. The extracted data encompassed study characteristics (including design, sample size and setting), participant demographics and comprehensive intervention details (such as type, duration and delivery format). Furthermore, we systematically extracted outcome measures, key findings, author affiliations (to specifically assess the extent of Palestinian researcher involvement), documented evidence of cultural adaptation and the stated theoretical framework. For studies that were excluded after full-text review, we conducted a thematic analysis to identify salient patterns across the literature. This analysis specifically focused on cataloging the types of interventions studied, critiquing the depth of cultural adaptation practices, quantifying Palestinian researcher involvement and synthesizing the apparent gaps in treatment approaches for this population.

### Results

#### Study selection

The study selection process is summarized in Figure 1. The primary search strategy focusing on narrative therapy (conducted through December 2023) yielded 847 potentially relevant records after removal of duplicates. Following title and abstract screening, 23 full-text articles were assessed for eligibility. After a detailed review, zero studies met the inclusion criteria for narrative therapy interventions with Palestinian populations.

**Table 1.** Search terms for systematic review

| Category | Search terms |
|---|---|
| Narrative therapy terms | "narrative therapy," "narrative practice," "narrative approach," "externalizing conversation," "unique outcome," "re-authoring," "counter-document," "narrative letter," "definitional ceremony," "outsider witness" |
| Palestinian populations | "Palestinian," "West Bank," "Gaza," "East Jerusalem," "refugee camp," "occupied territories," "Palestine," "conflict-affected Palestinian" |
| Mental health outcomes | "resilience," "mental health," "PTSD," "post-traumatic stress," "depression," "anxiety," "trauma," "psychological distress," "coping," "meaning-making," "hope," "agency," "quality of life," "sumud," "hikaye" |
| Broader intervention terms (secondary analysis) | "psychotherapy," "psychological intervention," "mental health intervention," "trauma therapy," "BT," "cognitive behavioral," "psychosocial support," "counseling," "treatment" |

*Note:* Primary search specifically excluded "narrative exposure therapy" and "NET" to maintain focus on White and Epston's narrative therapy framework. Secondary searches included broader intervention terms without exclusions.

The secondary broader search strategy, which replaced narrative therapy terms with general mental health intervention terms, yielded 1,203 records from databases and 34 from registers. After removing duplicates ($n = 390$), 847 unique records underwent title/abstract screening. The primary reasons for exclusion at this stage ($n = 824$) included non-Palestinian populations ($n = 412$), no mental health intervention component ($n = 278$), qualitative/observational studies without intervention ($n = 94$) and clearly irrelevant topics ($n = 40$). This process identified the same 23 full-text articles for detailed assessment, confirming that our narrative therapy-focused search had adequately captured the relevant intervention literature for Palestinian populations.

### Characteristics of excluded studies

Despite comprehensive database searches across multiple platforms, our review identified zero studies that specifically examined narrative therapy interventions with Palestinian populations. To map the broader evidence base and illustrate what types of mental health interventions have been studied with Palestinian populations, we provide a thematic overview of the 23 excluded full-text articles. This analysis demonstrates the rigor of our screening process and provides insight into patterns of research priorities, cultural adaptation practices and gaps in the intervention literature.

The 23 excluded studies fell into five distinct categories: CBT-based interventions with no narrative elements ($n = 11$); NET studies ($n = 4$); studies with non-Palestinian populations ($n = 4$); studies with no intervention component ($n = 3$); and narrative analysis only ($n = 1$) (see Table 2).

### Exemplary excluded studies and primary reasons for exclusion

#### Category 1: CBT-based interventions, no narrative elements ($n = 11$)
The majority of excluded studies ($n = 11$) employed CBT-based interventions focused primarily on symptom reduction without

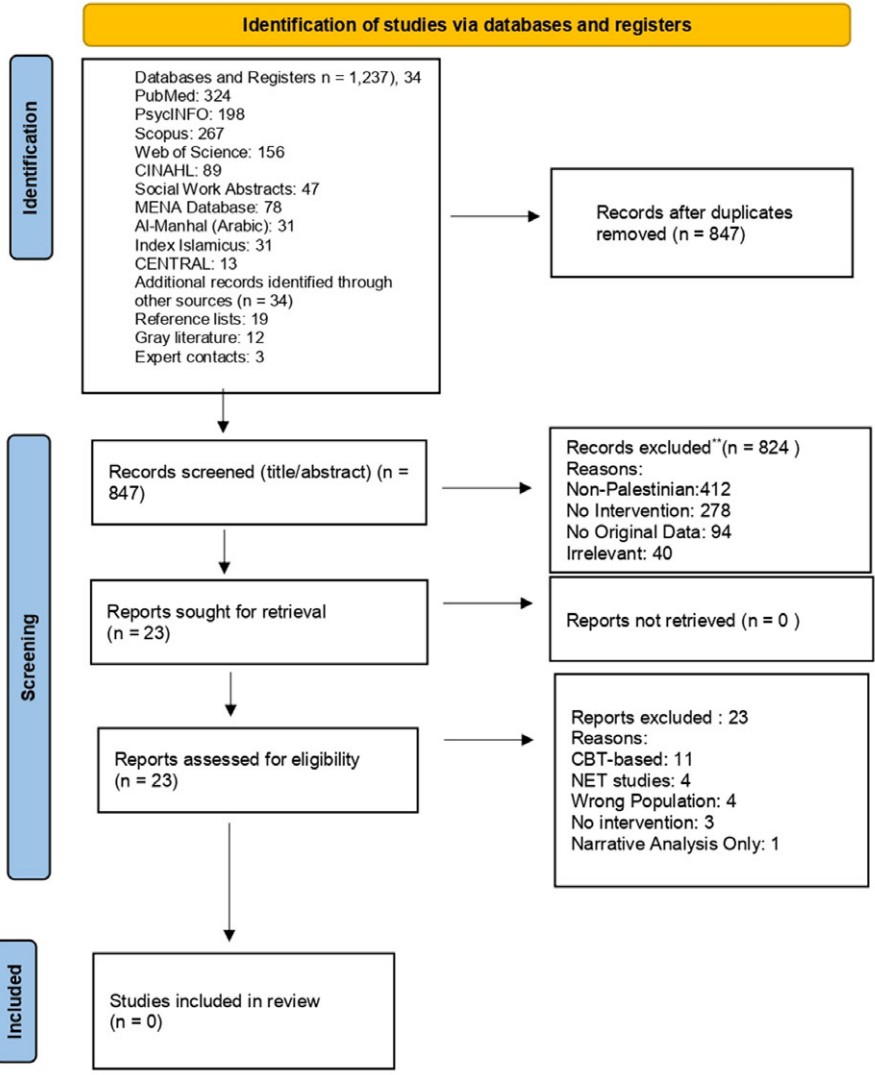

**Figure 1.** PRISMA 2020 flow diagram for new systematic reviews, adapted from Page MJ et al. BMJ 2021;372:n71. doi:10.1136/bmj.n71.

incorporating narrative therapy principles. Analysis of these studies revealed a pattern of utilizing standardized, Western-developed protocols, primarily TRT and trauma-focused CBT, with a mean duration of 7–12 sessions and a predominant focus on reducing symptoms of PTSD, depression and anxiety, demonstrating limited integration of Palestinian cultural frameworks. Regarding research leadership, only 5 of the 11 studies (45%) included Palestinian co-authors, with a mere 3 (27%) led by Palestinian principal investigators, though the majority (73%) involved international-Palestinian research partnerships. Cultural adaptation, when explicitly mentioned (which occurred in only four studies, or 36%), was typically superficial, focusing on language translation and contextual examples rather than modifying the theoretical framework; notably, only one study explicitly integrated the concept of *sumud*, and none systematically incorporated *hikaye* storytelling traditions. While these interventions were effective for their primary goal, all

studies showed significant symptom reduction with effect sizes ranging from small to large (Cohen's $d = 0.3–1.2$), their outcomes were limited by a lack of assessment on measures of cultural identity, collective resilience or community-level outcomes (see Table 3).

This analysis reveals that while CBT-based approaches demonstrate clear effectiveness for symptom management, their application within the Palestinian context is characterized by several significant limitations. These include a consistent pattern of limited theoretical diversity, an insufficient integration of Palestinian cultural frameworks and a predominance of Western-developed protocols. Furthermore, the research shows a constrained focus, often overlooking the profound dimensions of collective trauma and the ongoing political context, which is coupled with incomplete leadership from Palestinian researchers within the research process itself.

**Table 2.** Examples of excluded studies by category

| Study | Year | Intervention | Population | Primary exclusion reason |
|---|---|---|---|---|
| Barron, Abdallah and Smith | 2013 | Teaching Recovery Techniques (TRT) | Palestinian children, West Bank | CBT-based group intervention focused on trauma symptoms; no evidence of narrative therapy principles (externalization, unique outcomes and re-authoring) |
| Hussein et al. | 2020 | Narrative Exposure Therapy | Syrian refugees in Jordan | NET focuses on chronological trauma processing, not narrative therapy principles as defined by White and Epston; different theoretical foundation |
| Veronese et al. | 2012 | Psychosocial groups | General Middle Eastern children | General psychosocial support groups, not a narrative therapy approach; population not meeting Palestinian conflict-affected criteria |
| Marie, SaadAdeen and Battat | 2020 | Literature review | Palestinians | Systematic review of prevalence, no therapeutic intervention component |

*Note:* Table represents examples from each major exclusion category among the 23 full-text articles that underwent detailed evaluation.

**Table 3.** Category 1: CBT-based interventions, no narrative elements (*n* = 11)

| Study | Year | Intervention | Population | Findings | Exclusion reason | Concerns |
|---|---|---|---|---|---|---|
| Barron, Abdallah and Smith | 2013 | Teaching Recovery Techniques (TRT) | Palestinian children, West Bank | Significant improvements in PTSD, depression and grief symptoms through a 7-session CBT protocol | CBT-based group intervention focused on trauma symptoms, no externalization or unique outcomes exploration | No theoretical framework alignment with narrative therapy principles; standardized CBT approach |
| Diab, Palosaari, Punamäki et al. | 2015 | Teaching Recovery Techniques (TRT) | Palestinian children, Gaza | Enhanced resilience and reduced trauma symptoms in school-based delivery | CBT-based psychosocial intervention, no narrative therapy principles identified | Lack of narrative therapy methodology; focus on symptom reduction rather than story re-authoring |
| Qouta, Punamäki and El Sarraj | 2012 | TRT cluster intervention | Palestinian children, Gaza | Effective trauma symptom reduction in group format | Group CBT psychosocial program, no re-authoring or counter-documents used | No evidence of externalization practices or unique outcomes identification |
| Punamäki et al. | 2015 | Trauma-focused therapy | Palestinian mothers/ children | Improved maternal mental health and child adjustment | CBT approach focusing on trauma processing, not narrative identity work | Missing narrative therapy's emphasis on discourse deconstruction |
| Thabet and Vostanis | 2005 | Group crisis intervention | Palestinian children, Gaza | Reduced psychological distress through peer support | Crisis support groups, no therapeutic narrative work or discourse deconstruction | No systematic narrative practices were documented |
| El-Khodary et al. | 2020 | School-based trauma program | Palestinian students, Gaza | Decreased PTSD and depression symptoms in the school setting | CBT techniques and psychoeducation, no narrative therapy components | Absence of re-authoring processes or alternative story development |
| Khamis | 2005 | Psychosocial school program | Palestinian adolescents, Gaza | Enhanced coping skills and social support | Activity-based psychosocial support, no narrative component identified | No integration with Palestinian cultural narratives (*hikaye* and *sumud*) |
| Al-Krenawi et al. | 2011 | School counseling | Palestinian children, Negev | Improved academic and social functioning | General counseling and support, not a narrative therapy approach | Lack of systematic narrative methodology |
| Haj-Yahia et al. | 2013 | Community intervention | Palestinian adolescents | Strengthened community connections and coping | Community-based psychosocial support, no narrative-focused elements | No evidence of narrative identity work |
| Thabet et al. | 2000 | PTSD treatment | Palestinian youth | Reduced trauma symptoms through exposure techniques | Exposure therapy and CBT, not a narrative therapy methodology | Missing narrative therapy's focus on externalization |
| Jordans et al. | 2010 | Trauma intervention | Palestinian children | Improved mental health outcomes in a school setting | CBT-based intervention, no narrative approach elements | No systematic narrative practices |

## Category 2: Narrative exposure therapy studies, not narrative therapy (n = 4)

Four studies examined NET, which, despite the shared terminology, represents a fundamentally different therapeutic approach from the White and Epston model of narrative therapy. While NET has demonstrated effectiveness in various trauma-affected populations and centers on the construction of a chronological trauma narrative, its intervention characteristics, including a structured, exposure-based protocol focused on trauma memory consolidation and delivered in a standardized format, diverge significantly from narrative therapy. The core distinctions lie in NET's omission of key narrative therapy techniques; it does not employ externalization, focus on unique outcomes or engage in discourse deconstruction, reflecting its different theoretical underpinnings in exposure

**Table 4.** Category 2: Narrative exposure therapy studies, not narrative therapy (*n* = 4)

| Study | Year | Population | Intervention | Findings | Exclusion reason | Concerns |
|---|---|---|---|---|---|---|
| Hussein et al. | 2020 | Syrian refugees in Jordan | Narrative Exposure Therapy | Reduced PTSD symptoms through trauma narrative construction | NET focuses on chronological trauma processing, not narrative therapy principles | Different theoretical foundation; standardized trauma treatment vs. post-structuralist approach |
| Hijazi et al. | 2014 | Iraqi refugees | Group narrative intervention | Improved trauma symptoms in group format | Group NET approach, not a narrative therapy framework | Lacks externalization, unique outcomes and discourse deconstruction |
| Maalouf et al. | 2011 | Lebanese adolescents | School-based narrative program | Enhanced emotional regulation and coping | Narrative-based but not narrative therapy as defined by White and Epston | Missing core narrative therapy principles |
| International Trauma Center | 2007 | Palestinian students | School trauma program (CBI) | Reduced trauma symptoms in an educational setting | Classroom/Community/Camp-Based Intervention, structured program not utilizing NET or narrative therapy principles | No evidence of NET protocol (Schauer et al., 2011) or narrative therapy re-authoring; general psychosocial intervention |

therapy rather than a post-structuralist approach. Of the four NET studies identified, three were conducted with Syrian, Iraqi and Lebanese populations. One study (International Trauma Center, 2007) was conducted with Palestinian students but employed a classroom-based intervention program that, while addressing trauma, did not utilize NET methodology as defined by Schauer et al. (2011). Therefore, there remains an absence of rigorous NET clinical trials specifically with Palestinian populations using the established NET protocol (see Table 4).

This category reveals a critical gap in the literature: there is a complete absence of rigorous NET clinical trials using the established Schauer et al. (2011) protocol with Palestinian populations, despite demonstrated effectiveness in other Middle Eastern groups. This represents a significant missed opportunity to evaluate the efficacy of a structured, narrative-based approach, even one theoretically distinct from White and Epston's model, within this specific context. Furthermore, this gap underscores a broader need for research that examines whether therapeutic interventions centered on storytelling, a practice deeply embedded in Palestinian culture through traditions like *hikaye*, resonate with and are effective for these communities.

### Category 3: Not Palestinian population (n = 4)

Four studies were excluded as they did not focus on a Palestinian population as operationally defined for this review. The analysis revealed that these studies primarily examined Syrian, Iraqi, Lebanese or general "Arab" populations, with one study focusing on Palestinian citizens of Israel without establishing direct conflict exposure per our inclusion criteria. The interventions in these studies were varied, encompassing psychosocial groups, women's support groups and primary care counseling, though none employed a narrative therapy methodology. While these studies provide valuable insights into mental health in the broader MENA region, their findings are not directly transferable to the unique socio-political context of conflict-affected Palestinians living under occupation. This category ultimately highlights a significant literature gap, demonstrating that interventions tested in similar cultural or regional contexts have not been rigorously evaluated with the specific Palestinian populations that are the focus of this review (see Table 5).

The analysis of this category reveals two critical implications. First, it underscores the principle of contextual specificity: the mental health challenges and intervention needs of conflict-affected Palestinians are distinct, shaped by a unique matrix of prolonged occupation, displacement and political violence and, therefore, cannot be fully addressed by extrapolating from research on other populations, even within the broader Arab world. Second, this situation results in missed comparative insights; the absence of direct comparative studies between Palestinian and other regional populations limits the field's ability to disentangle universal mechanisms of therapeutic change from those that are uniquely effective within this specific socio-political context.

### Category 4: No intervention study (n = 3)

Three articles were excluded as they were not intervention studies but rather comprised a systematic review of mental health prevalence, a cross-sectional survey and a theoretical article. The analysis revealed that while these studies provide important contextual data for mapping the scope of the mental health burden and theoretical needs in Palestinian communities, they fundamentally lacked an interventional component. Their primary contribution lies in underscoring the high prevalence of mental distress, thereby highlighting a clear need for action; however, a key limitation is that they do not contribute any empirical evidence regarding the effectiveness of therapeutic treatments, which is the central focus of this systematic review (see Table 6).

The analysis of this category reveals a critical dissonance in the literature, which can be characterized by two interconnected themes. First, there is a clear abundance of documented need alongside a scarcity of evaluated solutions; the literature contains ample and compelling evidence of the profound mental health challenges faced by Palestinians, yet there remains a relative dearth of research rigorously evaluating concrete therapeutic interventions. Second, this highlights a significant theoretical-practical gap; while conceptual frameworks for intervention are proposed in the literature, they have largely not been translated into the empirical studies necessary to build a robust and applicable evidence base for effective treatment in this context.

### Category 5: Narrative analysis only (n = 1)

A single study was excluded because it conducted a narrative analysis without implementing a therapeutic intervention. The analysis revealed that this study employed qualitative narrative methodology to explore identity and cultural narratives among Palestinian youth, thereby providing rich, contextual data on

**Table 5.** Category 3: Not Palestinian population (*n* = 4)

| Study | Year | Population | Intervention | Findings | Exclusion reason | Concerns |
|---|---|---|---|---|---|---|
| Veronese et al. | 2012 | General Middle Eastern children | Psychosocial groups | Enhanced community resilience and social support | General psychosocial support, not narrative therapy; population not meeting Palestinian conflict-affected criteria | Population not specific to conflict-affected Palestinian context; intervention not narrative therapy |
| Shalhoub-Kevorkian | 2005 | Middle Eastern women | Women's group therapy | Improved empowerment and social connections | Support groups for women, not narrative therapy methodology | Non-Palestinian population; lack of a narrative therapy framework |
| Giacaman et al. | 2009 | General Arab adults | Primary care mental health | Enhanced mental health literacy and help-seeking | General primary care counseling, not narrative therapy; population criteria not met | Broad Arab population focus, not specific to Palestinians under occupation |
| Abo-Rass et al. | 2023 | Palestinian citizens of Israel | Mental health intervention | Improved access to culturally appropriate care | Palestinian population, but not in a conflict-affected context as operationally defined | The study focused on service utilization within Israel, not conflict-affected territories |

**Table 6.** Category: 4 No intervention study (*n* = 3)

| Study | Year | Type | Population | Findings | Exclusion reason | Concerns |
|---|---|---|---|---|---|---|
| Marie, SaadAdeen and Battat | 2020 | Literature review | Palestinians | High prevalence of mental health challenges documented | Systematic review of prevalence, no intervention research | No therapeutic intervention component; does not meet study design criteria |
| Gammoh et al. | 2024 | Cross-sectional study | Palestinian refugees in Jordan | Elevated rates of psychological distress were identified | Prevalence study, no intervention component | Descriptive study without an intervention; cannot assess therapeutic effectiveness |
| Mataria et al. | 2009 | Theoretical article | Palestinian communities | Conceptual framework for mental health intervention | Conceptual framework, no empirical intervention research | Theoretical article without empirical data or intervention evaluation |

**Table 7.** Category 5: Narrative analysis only (*n* = 1)

| Study | Year | Study type | Population | Findings | Exclusion reason | Concerns |
|---|---|---|---|---|---|---|
| Hammack, P. L. | 2021 | Narrative analysis | Palestinian youth, Gaza | Rich cultural narratives identified supporting identity development | Not an intervention study; theoretical/narrative analysis without therapeutic intervention | No therapeutic component; analytical rather than intervention research |

personal and collective stories. However, a crucial distinction from intervention research is that it *analyzed* narratives as its primary data, rather than *employing* narrative therapy techniques as a clinical intervention to facilitate therapeutic change. Consequently, this work falls squarely within the realm of descriptive, observational research and does not constitute the interventional science that is the focus of this review (see Table 7).

The analysis of this category yields two important insights. First, it provides a necessary methodological clarification, demonstrating that the presence of the term "narrative" in research does not equate to the clinical practice of Narrative Therapy, and highlights the distinction between narrative analysis as a research methodology and narrative therapy as a clinical intervention. Second, it underscores the value of such work as foundational research; studies of this nature provide a crucial understanding of the narrative ecology of Palestinian youth, which offers a rich, empirical foundation that could and should directly inform the culturally sensitive development and application of future narrative therapy interventions in this specific context.

### Summary of secondary analysis: Broader evidence base

Our secondary analysis of the 23 excluded studies reveals several concerning patterns in the broader evidence base for mental health interventions with Palestinian populations. First, we found a limited therapeutic diversity, with 11 of the 15 intervention studies (73%) employing CBT-based approaches and a complete absence of studies on psychodynamic, systemic or narrative therapeutic frameworks, alongside limited integration of culturally grounded healing practices. Second, there was insufficient Palestinian researcher involvement, with only 8 of the 23 total studies (35%) including Palestinian co-authors, a mere 3 (13%) led by Palestinian principal investigators and the majority of research conducted by international teams with limited equitable partnership. Third, cultural adaptation was often lacking or superficial; only 6 of the 15 intervention studies (40%) explicitly described adaptation processes, which were primarily limited to language and examples rather than theoretical modification, with minimal integration of key cultural assets like *sumud* or *hikaye* and no examination of community-

defined healing practices. Finally, our analysis identified critical missing treatment components, including a limited focus on collective trauma, minimal attention to the political context of ongoing oppression, rare assessment of cultural identity or community-level impacts and a complete absence of interventions designed to address meaning-making and resistance.

## Discussion

### Significance of the research gap

This systematic review identified a complete absence of published research on narrative therapy interventions within Palestinian communities, alongside a broader pattern of limited diversity and cultural grounding in the mental health intervention research base for Palestinian populations. Despite comprehensive searches across multiple databases, expert consultation and a gray literature review, we identified zero published studies examining narrative therapy as defined by White and Epston's (1990) framework in Palestinian contexts.

Our secondary analysis of the broader evidence base revealed additional concerning patterns: predominance of Western-developed CBT protocols, limited Palestinian researcher leadership, superficial cultural adaptation practices and insufficient attention to collective trauma, political context and culturally grounded healing frameworks.

### Interpreting the absence of narrative therapy research

While our stringent inclusion criteria were necessary to ensure fidelity to the White and Epston (1990) narrative therapy framework and to prevent conceptual confusion with other narrative-based approaches, this methodological rigor may have excluded interventions that incorporated some narrative therapy principles but did not meet our comprehensive criteria. However, the complete absence of studies meeting even basic narrative therapy criteria suggests that the research gap extends beyond methodological specificity to a fundamental absence of narrative therapy research in Palestinian contexts.

The absence of narrative therapy research must be interpreted alongside the broader evidence base. While CBT-based interventions have demonstrated effectiveness for symptom reduction, the lack of research on alternative approaches fundamentally limits our understanding in several key areas. It remains unknown whether different therapeutic frameworks might offer complementary or even superior benefits for specific populations or outcomes beyond symptom management. Furthermore, we lack evidence on how culturally aligned approaches might enhance participant engagement, treatment acceptability and the sustainability of long-term outcomes. Finally, this gap prevents us from investigating whether the deliberate integration of Palestinian cultural assets could potentially strengthen both established Western protocols and emerging indigenous therapeutic approaches, thereby creating more robust and resonant interventions.

### Theoretical-cultural compatibility and the value of diverse approaches

Universal biological stress responses mean that PTSD symptoms manifest similarly across contexts, and evidence-based trauma-focused interventions have demonstrated effectiveness for symptom reduction in diverse populations, including Palestinian

communities (Barron et al., 2013; Diab et al., 2015). These treatments address significant functional impairments and should be made available to Palestinian populations. CBT-based approaches like TRT have shown substantial value in reducing trauma symptoms and supporting recovery.

However, our analysis suggests that an exclusive focus on a single therapeutic paradigm, while valuable for symptom reduction, may limit critical opportunities for a more holistic form of healing. This narrow focus restricts the field's capacity to adequately address collective trauma and the political dimensions of distress, to integrate Palestinian cultural strengths and healing traditions, to support identity preservation and resistance to oppressive narratives and to foster meaning-making that extends beyond mere symptom reduction.

The question is not whether Western approaches should be excluded, but rather whether the mental health intervention landscape for Palestinian communities should be expanded to include complementary, culturally grounded approaches that may address dimensions of healing not fully captured by symptom-focused interventions.

### Decolonizing mental health practice

Evidence-based Western approaches have demonstrated clear value for symptom management and should remain accessible to Palestinian communities. However, the profound lack of diversity in the intervention research base underscores a significant and concerning gap in the literature, which must be contextualized within broader theoretical and structural frameworks. As Fanon (1963) argues in *The Wretched of the Earth*, colonial violence produces specific forms of psychological distress. Addressing this requires approaches that target both individual symptoms and structural oppression.

The complete absence of narrative therapy research, combined with limited diversity in the broader evidence base, underscores a critical gap in the decolonization of mental health approaches with Palestinian populations, a process called for by scholars such as Fanon (1963) and Tuhiwai Smith (2012).

While narrative therapy is itself a Western-developed approach, its post-structuralist, anti-oppressive theoretical framework and emphasis on deconstructing dominant discourses offer particular relevance for contexts of political oppression. Externalization practices could help Palestinians separate personal and collective identity from oppressive political narratives that systematically position them as inherently problematic or threatening, a process Said (1978) describes as orientalist discourse that dehumanizes Palestinian experience. This does not negate the value of CBT for symptom management, but suggests that additional therapeutic tools addressing discourse and identity may offer complementary benefits.

The narrative therapy focus on unique outcomes directly aligns with the Palestinian cultural concept of *sumud* (steadfastness), potentially highlighting acts of resistance, survival and cultural preservation often overlooked in deficit-focused trauma approaches (Halper, 2015). Current CBT-based interventions, while showing efficacy for symptom reduction, may miss opportunities to strengthen Palestinian cultural identity and collective resilience.

Re-authoring processes could support Palestinians in reconstructing their sense of identity and agency despite ongoing systematic displacement, restrictions and political violence, fostering empowerment rather than pathologization. This represents not a rejection of symptom-focused care, but an expansion of therapeutic

goals to encompass identity, meaning and resistance alongside symptom relief.

The principle of deconstructing dominant discourses directly addresses the need to critically examine colonizing narratives that contribute to psychological distress while positioning Palestinian experiences as secondary or illegitimate (Tuhiwai Smith, 2012). This critical consciousness development may be essential for healing in contexts of ongoing oppression and could complement rather than replace symptom-focused interventions.

### Expanding rather than replacing evidence-based approaches

Our analysis should not be interpreted as a dismissal of evidence-based Western therapeutic approaches or a suggestion that they be withheld from Palestinian communities. Rather, we advocate for expanding the evidence base to include rigorous evaluation of diverse therapeutic frameworks through culturally responsive, community-partnered methodologies. The extensive evidence for CBT-based trauma-focused interventions demonstrates their value for symptom reduction and functional improvement. Our critique focuses not on these approaches themselves, but on the systemic structures that have limited research on alternative or complementary therapeutic models.

Our primary concerns center on: (1) the systemic exclusion of alternative therapeutic models from research funding and academic attention; (2) the persistently limited integration of Palestinian cultural assets even within adapted Western frameworks; (3) the insufficient leadership from Palestinian researchers in defining and driving research priorities; and (4) the consequent missed opportunity to develop complementary approaches that might address essential dimensions of healing, such as identity, meaning and resistance, which lie beyond the scope of symptom reduction alone.

We strongly support making evidence-based trauma-focused interventions available to Palestinian communities through participatory research designs involving local researchers and affected communities. Simultaneously, we argue that the evidence base should be expanded rather than narrowed, allowing for evaluation of diverse therapeutic approaches that might offer complementary or culturally congruent benefits for dimensions of healing not fully addressed by symptom-focused interventions.

### Narrative therapy versus narrative exposure therapy: Clarifying cultural positioning

It is important to clarify that narrative therapy, similar to NET and CBT, is a Western-developed approach, originating in Australia and grounded in Western philosophical traditions (Foucault, Derrida and Bateson). We do not position it as "non-Western" in origin. However, we highlight: (1) the pronounced theoretical alignment between its principles and Palestinian cultural practices like *hikaye* and *sumud*; (2) its post-structuralist, anti-oppressive theoretical framework that resonates with decolonizing goals; and (3) its emphasis on deconstructing dominant discourses, which is highly relevant in a context of political oppression.

NET, with its focus on storytelling and dual aims of trauma processing and human rights documentation, also holds clear relevance for Palestinian contexts. The crucial distinction between the two approaches lies not in the value of storytelling, but in their differing theoretical foundations (post-structuralist identity work versus trauma exposure therapy), therapeutic processes (externalization and discourse deconstruction versus chronological trauma processing) and primary goals (identity re-authoring versus trauma memory consolidation). We contend that both approaches, among others, may offer distinct value and support future research that evaluates them through culturally responsive, community-partnered methodologies.

### Structural barriers and colonial biases in mental Health Research

The identified research gap reflects systemic barriers that extend beyond individual researcher interests or institutional capacity. These barriers illuminate the ways in which colonial structures continue to shape knowledge production in mental health, privileging Western paradigms and systematically excluding indigenous and culturally grounded approaches.

### Political and logistical challenges

Movement restrictions imposed by Israeli military occupation create significant logistical obstacles for conducting community-based intervention research (Abu-Lughod, 2007). International researchers face barriers to accessing Palestinian communities, while Palestinian researchers encounter restrictions on movement, institutional capacity and international collaboration. These conditions actively suppress the development of contextually relevant research.

### Funding biases and research colonialism

Funding mechanisms in international mental health research prioritize Western "evidence-based" approaches, creating a circular logic that excludes alternative models from evidence development while maintaining the dominance of Western therapeutic frameworks. This does not mean Western approaches lack value, but rather that the funding system creates self-fulfilling prophecies where only certain approaches receive validation through research. This represents a form of research colonialism where the very definition of "evidence" is shaped by Western academic and clinical paradigms. Major funding bodies consistently support research that replicates Western models rather than developing indigenous approaches, perpetuating what Tuhiwai Smith (2012) identifies as research colonialism.

Consequently, the current "evidence base" for interventions with Palestinian populations is systemically biased toward Western models, particularly CBT approaches, precisely because these are the interventions that receive funding. This creates a self-reinforcing cycle: Western frameworks are continuously validated through research, while culturally grounded approaches like narrative therapy remain marginalized and unstudied, falsely appearing to lack an evidence base.

### Institutional and epistemic barriers

Limited institutional capacity within Palestinian territories, combined with disrupted educational and healthcare systems due to ongoing conflict, further restricts opportunities for developing and evaluating innovative therapeutic approaches. Universities and training institutions lack resources and expertise in narrative therapy and diverse therapeutic modalities, while restrictions on international collaboration limit knowledge transfer. The academic publishing system itself constitutes a barrier, with journals prioritizing research from well-resourced Western institutions

and often lacking the cultural competency to evaluate innovative approaches developed in non-Western contexts.

### Implications for evidence-based practice

This systematic review challenges fundamental assumptions about evidence-based practice. The absence of research on narrative therapy does not indicate a lack of merit; rather, it reveals how research priorities and funding mechanisms systematically exclude culturally congruent interventions. Similarly, the predominance of CBT-based research, while demonstrating the value of these approaches for symptom management, may create a misleading impression that alternative approaches have been evaluated and found wanting, when in fact they have simply not been studied.

The extensive evidence base for CBT is valuable and should inform clinical practice. However, exclusive reliance on a single therapeutic paradigm may be insufficient for addressing the complex needs of communities experiencing ongoing political oppression and may inadvertently perpetuate colonial approaches that prioritize Western knowledge systems over indigenous wisdom. As our analysis of excluded studies demonstrates, the predominant focus on individual symptom reduction overlooks essential dimensions of collective healing, cultural preservation and resistance to oppressive structures. This bias reflects a deeper epistemological colonialism, where funding bodies and academic institutions privilege Western epistemologies while systematically excluding alternative frameworks that may be complementary, culturally compatible and therapeutically relevant for dimensions of healing beyond symptom reduction.

### Moving beyond Western frameworks while recognizing their value

Future intervention development should not abandon evidence-based Western approaches that have demonstrated effectiveness for symptom management, but rather should expand its priorities to create a more robust and culturally resonant evidence base. This entails a strategic shift in research toward: investigating how Palestinian cultural knowledge, practices and perspectives can enhance existing interventions; evaluating alternative therapeutic frameworks that may better address collective trauma, identity and meaning-making; examining whether the integration of cultural assets like *sumud* and *hikaye* strengthens treatment outcomes; and assessing whether diverse therapeutic approaches benefit different populations or address distinct dimensions of healing. This expansion necessitates a deeper engagement with Palestinian cultural wisdom, which includes recognizing storytelling traditions (*hikaye*) as sophisticated, sustained healing practices; understanding *sumud* as a complex framework encompassing resistance, hope, dignity and collective survival, far beyond Western conceptions of resilience; and acknowledging resistance narratives as vital sources of meaning-making and identity preservation. The ultimate goal is to examine how these assets can be authentically integrated to enrich both Western evidence-based approaches and culturally indigenous therapeutic frameworks.

### Conclusions and implications for researchers

#### The urgency of this research gap

Our systematic review documents a complete absence of published research on narrative therapy interventions with Palestinian populations, alongside broader patterns of limited therapeutic diversity, insufficient cultural adaptation and incomplete Palestinian researcher leadership in the mental health intervention research base. This represents both a significant missed opportunity and an urgent call to action. The theoretical promise of narrative therapy for Palestinian communities, combined with this complete absence of research and the broader pattern of limited therapeutic diversity, creates both a unique opportunity and a profound responsibility for the international research and clinical community.

The absence of evidence for narrative therapy, despite its alignment with Palestinian *sumud* (steadfastness) and oral storytelling traditions, reflects what we identify as a systematic bias in mental health research funding and a concerning neglect of culturally grounded interventions. This does not diminish the value of evidence-based Western approaches that have demonstrated effectiveness for symptom reduction, but suggests that the mental health research landscape for Palestinian populations would benefit from expanded diversity and cultural grounding. This gap perpetuates what Fanon (1963) and Tuhiwai Smith (2012) identify as colonial approaches to healing that may inadvertently reproduce oppressive relationships through therapeutic practice.

### Actionable recommendations

#### For researchers

For researchers, the imperative is to adopt participatory methodologies that fundamentally center Palestinian knowledge and leadership. This begins with establishing genuine collaborative partnerships that ensure community ownership of the research process, build local capacity and utilize community-based participatory research (CBPR) frameworks that honor Palestinian priorities. The research agenda itself must be expanded to critically and rigorously evaluate a diverse range of therapeutic approaches. This includes examining culturally aligned models like narrative therapy alongside evidence-based Western interventions, investigating deep cultural adaptation of existing treatments and assessing the complementary effectiveness of different frameworks. To achieve this, mixed-methods, community-led research designs are essential. These should involve sequential phases of partnership development, systematic cultural adaptation research (e.g., integrating *sumud* and *hikaye*), feasibility testing with culturally responsive measures and mixed-methods effectiveness trials that capture both symptom reduction and broader outcomes like cultural identity and collective resilience. Underpinning all efforts must be a critical examination of assumptions: avoiding the a priori exclusion of any approach based on its origin, and instead using rigorous evidence to determine cultural compatibility and clinical effectiveness for specific populations, contexts and therapeutic goals.

#### For funding bodies

A paradigm shift in funding priorities is essential to break the cycle of epistemic injustice and support genuine therapeutic diversity. Funding bodies must move beyond exclusive reliance on Western-centric models by actively prioritizing community-partnered grants that support Palestinian-led initiatives and provide long-term investment in local research infrastructure over short-term projects. This requires creating specific funding mechanisms for evaluating diverse therapeutic approaches, including narrative therapy and other culturally aligned interventions, comparative effectiveness research and studies that assess multiple dimensions of healing beyond symptom reduction. Crucially, these mechanisms must acknowledge and address structural barriers by accommodating political restrictions,

supporting collaboration despite access issues and prioritizing research on collective trauma and political dimensions of mental health. To break the self-reinforcing cycle where only prevalidated interventions receive funding, agencies must explicitly recognize that the absence of evidence is not evidence of absence. This involves providing seed funding for pilot studies of innovative, culturally grounded approaches and valuing diverse forms of evidence, including community wisdom and cultural practices, thereby creating a more equitable and effective global mental health research landscape.

### For universities and training institutions

Universities and training institutions bear a critical responsibility in preparing the next generation of clinicians and researchers. This requires the development of comprehensive training programs that move beyond single-paradigm education to emphasize diverse, culturally adapted therapeutic approaches. Curricula must be reformed to integrate decolonizing frameworks and critically examine the limitations and biases of dominant models. This involves providing rigorous training in multiple therapeutic frameworks, including narrative therapy and culturally adapted CBT, while systematically integrating Palestinian cultural assets like *sumud* and *hikaye* across modalities. Furthermore, training must extend beyond clinical technique to include CBPR methodologies, skill-building in critical consciousness and anti-oppressive practice and a thorough understanding of the structural barriers in global mental health. To support this, institutions should actively foster Palestinian-led scholarship, establish equitable partnerships with Palestinian educational institutions and create exchange programs that facilitate genuine knowledge co-creation, thereby transforming educational ecosystems to be more just, relevant and effective.

### For Palestinian communities

Ultimately, the most transformative shift requires that Palestinian communities transition from being subjects of research to leaders of their own healing inquiry. This entails communities authentically leading research priorities by building local capacity for therapeutic innovation, preserving and centering their own healing wisdom and defining what constitutes "effective" intervention beyond Western symptom measures – all based on community-identified needs rather than external agendas. Concurrently, communities are essential contributors to knowledge preservation and development; this includes documenting traditional practices, training community members in research methodologies, creating networks for knowledge sharing and advocating for community-controlled research initiatives. By maintaining ownership of the knowledge and research processes that concern them, Palestinian communities can ensure that mental health interventions are not only culturally resonant but also empowering and self-determined.

### For clinicians and policymakers

For these research advancements to translate into meaningful change, parallel shifts are required in clinical practice and policy. Clinicians must operate from a principle of inclusive evidence-based practice, which involves making proven trauma-focused interventions available while simultaneously exploring and evaluating culturally grounded approaches. This includes actively integrating Palestinian cultural assets into treatment and assessing multidimensional well-being, encompassing symptoms, cultural identity, meaning-making and empowerment, while recognizing that the absence of evidence for an approach does not equate to evidence of ineffectiveness. Correspondingly, policy must actively

support this evolution by mandating therapeutic diversity in service provision, funding pilot programs for innovative and culturally grounded approaches, requiring Palestinian leadership in all research and program development and institutionalizing cultural adaptation as a mandatory process for all mental health interventions.

### Future research directions

This systematic review identifies several critical and interconnected pathways for future research to address the identified gaps. The immediate agenda must include conceptual work to develop frameworks that map the alignment between diverse therapeutic principles and Palestinian cultural assets, and comparative effectiveness research that evaluates narrative therapy, culturally adapted CBT, NET and other approaches to determine their respective benefits for different populations and dimensions of healing. Advancing this agenda necessitates methodological innovation to create measures that capture collective resilience and community empowerment, and dedicated adaptation research to explore the authentic integration of cultural assets like *sumud* and *hikaye* across therapeutic modalities. Finally, implementation science is needed to develop delivery strategies for the unique constraints of the Palestinian context, and new training models must be created to prepare clinicians in these culturally adapted methods.

The implications of addressing this research gap extend far beyond the Palestinian context. By tackling the dual challenge of a specific evidence absence and a broader lack of therapeutic diversity, this work has the potential to make seminal contributions to global mental health. It can advance our understanding of culturally responsive interventions in conflict settings, decolonizing therapeutic practices that center on indigenous knowledge and holistic healing models that integrate social justice. Ultimately, research in this domain can demonstrate how to support not only individual symptom relief but also collective resilience, cultural preservation and community empowerment in the face of systematic oppression, thereby contributing to both Palestinian well-being and the global knowledge base on community-led mental health intervention.

### Strengths and limitations

This review possesses several notable strengths that bolster the credibility and impact of its findings. These include its systematic, preregistered methodology conducted in strict adherence to PRISMA guidelines, a comprehensive search strategy executed across multiple databases and languages and rigorous, multistage screening and data extraction processes. Furthermore, a significant strength lies in its secondary analysis, which moves beyond a simple "empty review" to map the broader evidence base, thereby providing a critical examination of Palestinian researcher involvement, cultural adaptation practices and the structural barriers and research biases that shape the current literature.

This review has several limitations to consider. While our stringent inclusion criteria ensured fidelity to the White and Epston (1990) narrative therapy framework and prevented conceptual confusion, this rigor may have excluded interventions incorporating some narrative principles without meeting all our criteria. However, this was necessary to answer our specific research question, and the fact that no studies met even the basic criteria confirms a fundamental absence of narrative therapy research, not just a methodological artifact. Additional limitations include the potential for publication bias (though it cannot explain a total absence), the restriction to English, Arabic and Hebrew languages and the

possibility that our comprehensive gray literature search still missed unpublished pilots. Furthermore, our secondary analysis of excluded studies was post-hoc, and we had a limited ability to assess the quality of cultural adaptation based solely on published descriptions. Most importantly, our critique of limited therapeutic diversity should be interpreted not as a dismissal of evidence-based Western approaches, but as a call for expanded research on diverse frameworks through rigorous, culturally responsive methodologies.

## Final call to action

The complete absence of narrative therapy research with Palestinian populations, combined with broader patterns of limited therapeutic diversity and cultural grounding in the mental health intervention research base, represents a significant and concerning gap in the literature, given the well-documented theoretical alignment and acute mental health needs. The theoretical compatibility between narrative therapy principles and Palestinian cultural assets creates an unprecedented opportunity to address the urgent mental health needs of Palestinian communities through meaningful, culturally grounded intervention development alongside continued evaluation and cultural adaptation of evidence-based Western approaches.

Consequently, addressing the critical gaps identified in this review demands a concerted and collaborative effort from researchers, funding agencies, academic institutions and Palestinian communities. This collective action must be directed toward a clear agenda: evaluating narrative therapy and other culturally aligned interventions through rigorous research; expanding therapeutic diversity in mental health services; enhancing the cultural adaptation of existing evidence-based interventions; and ensuring Palestinian leadership in all stages of research and implementation. Underpinning these actions must be a commitment to addressing the structural barriers and funding biases that have historically limited research on culturally grounded approaches. It is vital to frame this endeavor not as a rejection of effective Western models, but as a necessary enrichment of the evidence base to provide a diverse, culturally responsive and comprehensive spectrum of healing pathways for Palestinian communities.

Supporting community-partnered studies that center Palestinian voices and knowledge systems is a necessary next step. Such research is vital for developing effective interventions that are not only culturally grounded but also empower community-defined healing and recovery. The goal is not to exclude any therapeutic approach, but to ensure that Palestinian communities have access to the full range of potentially beneficial interventions, evaluated through rigorous research that honors both universal principles of effective treatment and culturally specific pathways to healing.

**Open peer review.** To view the open peer review materials for this article, please visit http://doi.org/10.1017/gmh.2025.10103.

**Data availability statement.** No primary data were generated or analyzed in this study. The systematic review protocol is registered on PROSPERO (CRD420251128423). All search strategies and the list of excluded studies are available within the article.

**Acknowledgments.** The authors would like to express their profound gratitude to the Deanship of Applied Research at Nablus University for Vocational and Technical Education for their continuous and invaluable support. The authors would also like to extend our sincere thanks to the librarians at Nablus University for Vocational and Technical Education for their indispensable assistance in developing the comprehensive search strategy.

The authors are deeply grateful to Professor Johanne Goudreau of the University of Montreal for her invaluable guidance and support throughout this project. The authors would also like to thank the peer reviewers whose thoughtful comments significantly strengthened this manuscript and challenged the authors to clarify their position on the complementary value of diverse therapeutic approaches and the importance of evidence-based interventions in conflict-affected contexts.

**Author contribution.** I.A.: Conceptualization, methodology, formal analysis, investigation, writing – original draft, writing – review and editing and project administration. M.S.: Conceptualization, methodology, validation, writing – review and editing and supervision. S.A.: Investigation, data curation and writing – review and editing. K.S.: Investigation, data curation and writing – review and editing.

**Financial support.** This research received no specific grant from any funding agency, commercial or not-for-profit sectors.

**Competing interests.** The authors declare none.

**Ethics statement.** As a systematic review of published literature, this study did not require ethical approval or informed consent.

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
