## [Reviewer Report]

Dear Authors,

the manuscript titled “Narrative Therapy Interventions for Resilience and Mental Health in Conflict-Affected Palestinian Communities: An Empty Systematic Review.” This is a rigorously conducted and exceptionally well-written review that addresses a critical and underexplored gap in the literature. The manuscript makes a significant contribution by highlighting a systemic bias in mental health research and proposing a decolonizing path forward. The methodology is sound, and the discussion is theoretically rich. Below are important suggestions to further strengthen an already excellent manuscript.

Abstract: The abstract is clear and concise. To enhance its impact, consider adding a sentence to explicitly state the primary implication of finding zero studies (e.g., “The absence of evidence, despite strong theoretical alignment, points to a systemic funding and research bias against culturally congruent interventions.”).

Introduction: The introduction is comprehensive and compelling. One minor addition could be to briefly define “empty review” early on, perhaps in the rationale section, to immediately frame the methodological approach for the reader, even though it is detailed later.

Methods: The methodology is robust and transparently reported.

-The PRISMA flow diagram is excellent. For even greater clarity in the main text, you might briefly note the primary reasons for exclusion at the title/abstract stage (n=824) in a footnote or parenthesis, as you do so well for the full-text exclusions.

-While the exclusion of NET is well-justified, consider adding a single sentence in the exclusion criteria section reiterating that NET was excluded due to its different theoretical foundation (trauma processing vs. post-structuralist identity work) to preempt any reader confusion.

Results: The results are presented clearly. The tables categorizing excluded studies are particularly valuable.

-In Table 2 (“Exemplary Excluded Studies”), the column “Primary Exclusion Reason” is very effective. For consistency, consider adding this same column to the more detailed tables (Category 1-5) to allow for quick scanning.

-In the “Category 4: No Intervention Study” table, the “Concerns” column for the Marie et al. (2020) entry states “No therapeutic intervention component.” This is accurate, but the “Concerns” column in other tables lists methodological/theoretical mismatches. For consistency, a phrase like “Does not meet intervention study criteria” might be slightly more precise in this context.

Discussion: The discussion is a major strength of the manuscript, thought-provoking, critical, and forward-looking.

-The section on “Structural Barriers and Colonial Biases” is powerful. To make the argument about funding even more concrete, you could consider adding a sentence speculating on the mechanism (e.g., “Funding bodies may perceive Western models like CBT as lower-risk investments due to their existing evidence base, creating a cycle that systematically disadvantages the development of evidence for culturally-nuanced approaches like narrative therapy.”).

-The “Actionable Recommendations” section is outstanding and a model for empty reviews. To enhance it further, you might briefly mention the importance of publishing negative or null results from future pilot studies to contribute to the evidence base and avoid publication bias.

Notes to consider: The writing is clear, engaging, and academically sophisticated. Minor points: Some acronyms are defined multiple times (e.g., NET in the abstract, introduction, and methods). While this is sometimes done for reader convenience, consider consistency (e.g., defining it once in the introduction and then using the acronym thereafter).

References: The reference list is comprehensive and appropriate. The use of culturally relevant and critical theory sources (Fanon, Tuhiwai Smith, Said) is a particular strength that solidifies the manuscript’s decolonial argument.

---

## [Reviewer Report]

The paper “Narrative Therapy Interventions for Resilience and Mental Health in Conflict-Affected Palestinian Communities: An Empty Systematic Review“ investigates if and how Narrative Therapy has been investigated in research papers as an intervention for Palestinian communities. The authors conducted a throughout systematic review following highest standards (e.g. preregistration, systematic protocol).

All in all, the scientific approach of the review is very sound. However, I see limitations in the conceptualization and interpretation of the review:

1. Implications for Western-derived therapeutic approaches

The authors argue that Narrative Therapy is culturally aligned with Palestinian traditions, while other approaches—particularly CBT-based interventions such as Narrative Exposure Therapy (NET)—are portrayed as Western concepts that do not fit the local context. This argument is embedded within a critique of colonial legacies in research and mental health care, which I strongly acknowledge as highly relevant. Nonetheless, I see the risk that this line of reasoning could lead to a generalized dismissal of Western approaches and thereby inadvertently limit access to effective treatments in Palestinian communities.

• Extensive evidence shows that symptoms of Posttraumatic Stress Disorder (PTSD) manifest in broadly similar ways across different contexts, which is consistent with the universal biological stress response underlying trauma.

• Untreated PTSD can lead to profound, long-lasting functional impairments (e.g., inability to work or maintain close social relationships).

• Consequently, evidence-based trauma-focused interventions should not be excluded a priori but rather critically examined within the local context. Ideally, this would be achieved through participatory research designs involving local researchers and affected communities, ensuring both contextual adaptation and empirical validation.

2. Ambiguity in the cultural positioning of Narrative Therapy

The review frames Narrative Therapy as a non-Western approach. However, it is important to note that Narrative Therapy was originally developed by Western clinicians (Epston & White). While its emphasis on storytelling may indeed resonate with Palestinian cultural practices, a similar argument can be made for Narrative Exposure Therapy (NET), which also centers around storytelling. Importantly, NET was explicitly developed for survivors of multiple and systematic violence in diverse global contexts, with the dual aim of empowering individuals to tell their stories and of documenting human rights violations. These aims could also be highly relevant in the Palestinian context.

3. Recommendation for a broader scope

Rather than focusing narrowly on Narrative Therapy while excluding other approaches a priori, the review could provide greater value by mapping the broader evidence base on mental health interventions in Palestine. This might include:

• Examining the extent of Palestinian researcher involvement in published studies.

• Identifying if and how interventions were adapted to the contexts of Palestinian communities.

• Highlighting which treatment components may still be missing or underdeveloped in this context.

Such an approach would allow for a more nuanced and comprehensive understanding of mental health care in Palestine. Importantly, it would avoid the implication that certain therapeutic concepts are inherently unsuitable for this context, while also safeguarding against the risk of withholding effective interventions and resources from Non-Western communities.

---

## [Editor Report]

Dear authors,

Thank you very much for submitting this paper to Cambridge Prisms: Global Mental Health. 

Please find herewith reviewers' responses for your submission. We would like to invite you to make major revisions to the paper. From an editorial standpoint, we believe it is important to make changes to the paper in line with reviewer 2’s comment: “Rather than focusing narrowly on Narrative Therapy while excluding other approaches a priori, the review could provide greater value by mapping the broader evidence base on mental health interventions in Palestine.”

---

## [Reviewer Report]

I would like to thank the authors for submitting the revised version of the manuscript and for their careful consideration of the previous comments.

I have the following remaining remarks regarding the revised version:

1. I appreciate that the authors have expanded their search strategy to include a broader range of evidence on mental health interventions (search strategies: broader intervention terms, p.7). However, this does not appear to be reflected in the results section or the flow chart, which remain identical to those presented in the previous version of the manuscript. Could the authors please elaborate on this point? In the current version, it appears that the secondary analysis was conducted on the excluded papers, while the main focus of the search strategy continued to center on narrative therapy (as all search strategies included terms related to narrative therapy and the Boolean operator AND). How can the authors ensure that the review comprehensively captures the full range of mental health interventions relevant to the Palestinian context?

2. I also appreciate the authors’ acknowledgment of the importance of implementing evidence-based (trauma-focused) therapeutic approaches in the Palestinian context, given its long history of exposure to violence. However, this point has been mentioned only in the discussion section and not in the introduction. I believe it would be important to acknowledge this perspective already in the introduction to provide appropriate context for the reader.

3. Furthermore, the authors have agreed with my previous comment that Narrative Therapy is also a Western approach. However, this is mentioned only in the discussion section. I would recommend including this information in the introduction when Narrative Therapy is first introduced, to provide a more balanced and critical framing.

4. The authors state that there is a complete absence of NET studies in the Palestinian context; however, Table 2 lists one study conducted with Palestinian students. This inconsistency should be clarified.

5. In the discussion section, the authors make direct references to reviewer comments. As a result, parts of the discussion read somewhat like a response letter. I would recommend rephrasing these sections to ensure that the discussion maintains an academic and narrative tone consistent with the rest of the manuscript.

---

## [Editor Report]

Dear Authors, thank you very much for your revision of the manuscript. Could you please take a close look at the comments from the second reviewer? Best wishes, Wietse

---

## [Reviewer Report]

Thank you very much for your thoughtful revision of the manuscript.

I have one minor point (that can be adressed by the authors without any further external review): I think in the revised abstract, Results section it should read “The included literature” instead of “The excluded literature”.

---

## [Editor Report]

Dear authors,

Thank you for the revision. Could you please take a look at the final minor comment from the reviewer, and then we would be glad to accept the paper.

---

## [Editor Report]

Dear authors,

Thank you very much for the additional revision. We are pleased to accept the manuscript.